# Identification of the Regulatory Targets of miR-3687 and miR-4417 in Prostate Cancer Cells Using a Proteomics Approach

**DOI:** 10.3390/ijms231810565

**Published:** 2022-09-12

**Authors:** Simone Venz, Heike Junker, Erik Ultsch, Franziska Hetke, Elke Krüger, Martin Burchardt, Pedro Caetano-Pinto, Cindy Roennau

**Affiliations:** 1Department of Medical Biochemistry and Molecular Biology, University Medicine Greifswald, 17475 Greifswald, Germany; 2Department of Urology, University Medicine Greifswald, 17475 Greifswald, Germany

**Keywords:** microRNA, prostate cancer, castration-resistant prostate cancer, cellular regulation, proteomics

## Abstract

MicroRNAs (miRNA) are ubiquitous non-coding RNAs that have a prominent role in cellular regulation. The expression of many miRNAs is often found deregulated in prostate cancer (PCa) and castration-resistant prostate cancer (CRPC). Although their expression can be associated with PCa and CRPC, their functions and regulatory activity in cancer development are poorly understood. In this study, we used different proteomics tools to analyze the activity of hsa-miR-3687-3p (miR-3687) and hsa-miR-4417-3p (miR-4417), two miRNAs upregulated in CRPC. PCa and CRPC cell lines were transfected with miR-3687 or miR-4417 to overexpress the miRNAs. Cell lysates were analyzed using 2D gel electrophoresis and proteins were subsequently identified using mass spectrometry (Maldi-MS/MS). A whole cell lysate, without 2D-gel separation, was analyzed by ESI-MS/MS. The expression of deregulated proteins found across both methods was further investigated using Western blotting. Gene ontology and cellular process network analysis determined that miR-3687 and miR-4417 are involved in diverse regulatory mechanisms that support the CRPC phenotype, including metabolism and inflammation. Moreover, both miRNAs are associated with extracellular vesicles, which point toward a secretory mechanism. The tumor protein D52 isoform 1 (TD52-IF1), which regulates neuroendocrine trans-differentiation, was found to be substantially deregulated in androgen-insensitive cells by both miR-3687 and miR-4417. These findings show that these miRNAs potentially support the CRPC by truncating the TD52-IF1 expression after the onset of androgen resistance.

## 1. Introduction

Prostate cancer (PCa) is the most diagnosed malignant disease in men and is still the second leading cause of cancer-related deaths in the Western world [1,2,3]. Radical prostatectomy (RPE) or radiotherapy are effective treatment options for clinically localized diseases. The treatment for metastatic PCa is based on androgen ablation. Upon androgen deprivation therapy (ADT), the disease may further progress to castration-resistant prostate cancer (CRPC), which has a much poorer prognosis [4]. CRPC can manifest differently, from patients without any evidence of metastases or symptoms to patients with bone, lymph node, and visceral metastases and experiencing cancer-related symptoms. The median time to develop CRPC ranges from 12 to 23 months, and the median time from diagnosis to mortality in CRPC patients ranges from 23 to 37 months [5,6].

According to the European Association of Urology (EAU) 2021 guidelines, CRPC is defined by castrate serum testosterone < 50 ng/dL (or 1.7 nmol/L), in conjunction with either biochemical progression (three consecutive rises in PSA one week apart, resulting in two 50% increases over the basal levels, with PSA > 2 ng/mL) or radiological progression (the appearance of two or more new bone lesions on a bone scan or the enlargement of a soft tissue lesion using Response Evaluation Criteria in Solid Tumors) [7]. Although the advances in the treatment of CRPC have improved the overall survival and quality of life, there are still no curative therapy options available [8]. A better understanding of the pathogenesis of CRPC is required to develop new treatment and diagnosis modalities. Extensive studies have been conducted to investigate the pathogenesis of PCa and the molecular mechanisms behind the transition to castration resistance [9]. Research has focused primarily on the functional deregulation of the androgen receptor (AR); AR signaling is pivotal in the development of potent ADT drugs, such as abiraterone acetate and enzalutamide, for the treatment of PCa patients [10].

MicroRNAs (miRNAs) are small non-coding, single-stranded RNA molecules, usually consisting of 17 to 25 nucleotides in length, that regulate gene expression at the post-transcriptional level [11,12]. The expression of these oligonucleotides is extensively reported across several cancers and is reported to be involved in tumor pathogenesis and progression [12,13,14]. miRNAs modulate the expression of their target genes, in general, by binding to the 3′ untranslated region (UTR) of target mRNAs, leading to either degradation or inhibiting translation; thus, effectively suppressing the expression of specific proteins [13]. More than 2600 human miRNA sequences have been identified so far (miRBase release 21; Genome Reference Consortium) [14]. Individual miRNAs are predicted to regulate the expression of multiple mRNAs and, likewise, a given mRNA strand can be targeted by multiple and different miRNAs [15]. An intricate regulatory mechanism miRNA from the same family are expected to have overlapping targets. In cancer, the miRNA-mediated dysregulation of protein-coding genes is involved in important cellular processes governing angiogenesis, apoptosis, cell proliferation, differentiation, and invasion [11,16,17,18]. MiRNAs can act as oncogenes (oncomiRs) or as tumor suppressors, depending on cell type-specific expression and activity [17,19].

Our understanding of the dysregulation of miRNAs in CRPC and their functional role in the development and progression to locally advanced, metastatic, and finally, castration-resistant prostate cancer, is still limited [20]. Most interestingly, miRNAs are associated with the AR, regulating the receptor expression at the mRNA or protein level [21,22]. Several differentially expressed miRNAs have been identified in CRPC tissue or in vitro models based on microarray analysis or deep-sequencing data [23]. Androgen-regulated miRNAs, including miR-21, miR-32, miR-99a, miR-99b, miR-148a, miR-221 and miR-590-5p, were reported to be differentially expressed in CRPC compared to benign prostate hyperplasia based on microarray analysis [24]. Functional studies have shed light on the role of miRNA in CRPC. Notably, the effects of miR-34a [25] and miR-19a [26] expressions were reported to be significantly decreased or increased in human PCa specimens, respectively. The overexpression of miR-34a in vitro resulted in the reduced expression of AR, PSA, and NOTCH-1, as well as the inhibition of the growth of PCa cells [25]. Inhibition of miR-19a in CRPC cells downregulated the expression of the tumor suppressor gene BTG1—B-cell translocation gene—and, subsequently led to the suppression of cell proliferation and increased apoptosis [26].

The identification of CRPC-associated miRNAs leads to a better understanding of the molecular alterations associated with the pathogenesis of PCa and the progression to CRPC. In a recent profiling study comparing the expressions of several miRNAs in CRPC and primary PCa tissue, we found that miR-3687 and miR-4417 were the most consistently upregulated miRNAs in CRPC across all the specimens analyzed [27]. In the present study, we employed a proteomics approach to identify the regulatory role of miR-3687 and miR-4417, as well as their target proteins in prostate cancer. Unraveling the regulatory networks and protein targets of CRPC-specific miRNA is important to further understand their impact on prostate cancer, particularly their role in the transition to androgen resistance [28]. The introduction of new and more effective drug modalities such as antisense oligonucleotides (AON) and proteolytic-targeting chimeras (PROTAC) requires the identification of key regulatory disease targets [29,30]. AON and PROTACS can be designed to exclusively target a specific RNA sequence or a protein of interest, respectively. Therefore, characterizing the proteins regulated by CRPC-specific miRNA and their functional effects is crucial for the development of new therapies.

## 2. Results

### 2.1. miR-3687 and miR-4417 Are Highly Overexpressed after Transfection in Prostate Cancer Cells

Transient transfection of miR-3687 or miR-4417 in the prostate cancer cells used yielded a substantial increase in the expression of both miRNAs (Figure 1). Relative to un-transfected cells (native expression), miR-3687 levels were upregulated by 1984-fold in PC-3, 1230-fold in LNCaP, 1697-fold in CTM-LNCaP, and 7727-fold in VCaP. miR-4417 levels were upregulated by 536-fold in PC-3, 841-fold in LNCaP, 1343-fold in CTM-LNCaP, and 1214-fold in VCaP. These results demonstrate the efficacy of transfecting both miR-3687 and miR-4417 into prostate cancer cells. The extensive upregulation in expression achieved is required to promote a measurable impact of both miRNAs in the physiology of the prostate cancer cells.

### 2.2. Proteome Analysis of Prostate Cancer Cell Lines after Transfection with miRNA

#### 2.2.1. Differentially Expressed Proteins in Cancer Cell Lines after miR-3687 Overexpression—2D Gel Electrophoresis

Two-dimensional gel electrophoresis determined that miR-3687 also yielded a substantial degree of deregulation across all prostate cancer cell lines analyzed (Appendix A). Only spots and proteins identified in all three biological replicates were included in our analysis. In Figure 2, the quantitative comparisons of the protein spots of the respective replicates for the cell lines are visualized using heat maps, and the principal component analysis (PCA) plot statements are presented as evidence of the differentiation of the samples for the individual cell lines. In PC-3 (Figure 2A) cells, a total of 13 spots were identified, comprising 27 proteins relative to the transfection control. In LNCaP (Figure 2B), 22 spots corresponded to 64 proteins; in CTM-LNCaP (Figure 2C), 7 spots corresponded to 14 proteins; and in VCaP (Figure 2D) cells, 33 spots comprising 83 proteins were detected.

#### 2.2.2. Differentially Expressed Proteins in Cancer Cell Lines after miR-4417 Overexpression—2D Gel Electrophoresis

Two-dimensional gel electrophoresis was used to separate proteins and identify proteins whose expression was altered after miR-4417 overexpression in four different prostate cancer cell lines. The gels obtained and the subsequent expression analysis can be found in Appendix B. Overall, miR-4417 yielded a substantial degree of deregulation across all cell lines. As already explained for the miR-3687, heatmaps and PCA plots were also generated for the analysis of the samples with the miR-4417 (Figure 3). Only spots and proteins identified in all three biological replicates were included in our analysis. In PC-3 (Figure 3A) cells, a total of 11 spots were differentially expressed, and were identified comprising 34 proteins. In LNCaP (Figure 3B), 28 spots correspond to 56 proteins; in CTM-LNCaP (Figure 3C), 16 spots correspond to 53 proteins; and in VCaP (Figure 3D) cells, 25 spots comprising 66 proteins were detected.

#### 2.2.3. Proteome Analysis of Prostate Cancer Cell Lines after miR-3687 Overexpression—Gel-Free Approach

Global proteomic analysis retrieved a total of 255 proteins quantifiable across the four cell lines overexpressing miR-3687 and transfection controls in gel-free samples. Only proteins detected in all three biological replicates were included in the analysis. A list of proteins identified and their expression ratios relative to the transfection controls (no microRNA overexpression), which were deregulated in at least two cell lines, can be found in Appendix C, Table A1. The majority of deregulated proteins were exclusively expressed in each cell line (LNCaP: 38 out of 72; CTM-LNCaP: 38 out of 73; PC-3: 105 out of 137; VCaP: 22 out of 35), as depicted in Figure 4A.

#### 2.2.4. Proteome Analysis of Prostate Cancer Cell Lines after miR-4417 Overexpression—Gel-Free Approach

Global proteomic analysis retrieved a total of 262 proteins quantifiable across the four cell lines overexpressing miR-4417 and the respective controls in gel-free samples. Only proteins detected in all three biological replicates were included in the analysis. The list of proteins identified and their expression ratios relative to the transfection controls, which were deregulated in at least two cell lines, can be found in Appendix C, Table A2. A total of 40 are commonly expressed by all cell lines, as depicted in Figure 4B. Individually, LNCaP exclusively expresses 26 out of 53 deregulated proteins; CTM-LNCaP 31 out of 52; PC-3: 137 out of 163; VCaP: 28 out of 37.

### 2.3. Functional Enrichment Analysis of Differentially Expressed Proteins after miRNA Transfection

#### 2.3.1. Profiling after miR-3687 Overexpression

Functional enrichment analysis was performed using the web server g:Profiler [31] to retrieve the ontology of the proteins identified by proteomic analysis (Figure 5A,C); both proteins isolated the identified spots in 2D gels and gel-free samples. The protein ontologies determined are described in Table 1.

The functions determined for miR-3687 differ from the 2D-gel and gel-free analysis. The most significant ontologies determined for the 2D-gel samples were related to the molecular function of the redox proteins involved in the biological responses to inorganic substances and the association with myosin fibers. In the gel-free analysis, the proteins deregulated by miR-3687 were determined to have molecular functions related to RNA, and protein binding involved in the biological processes of cytoplasmic translation and the biosynthesis of macromolecules. The cellular compartments associated with the proteins were identified as extracellular membrane-bound organelles and extracellular endosomes.

#### 2.3.2. Profiling after miR-4417 Overexpression

Functional enrichment analysis determined that 2D-gel identified proteins are involved in RNA and organic compound binding, and their biological activities are responsive to cellular stress while associated with organelles and extracellular vesicles. In the gel-free analysis, the proteins deregulated by miR-4417 were found to be associated with RNA and protein binding related to the biological processes of cytoplasmic translation and the biosynthesis of macromolecules, as well as being associated with extracellular membrane-bound organelles and extracellular endosomes. The proteins identified in the 2D-gel and gel-free approaches have consistent biological functions (Figure 5B,D). The protein ontologies determined are described in Table 1.

In the gel-free approach, the proteins deregulated by both miR-3687 and miR-4417 share the same oncology. On the other hand, the 2D-gel approach retrieved stark functional differences between both miRNAs. This is likely due to the fact that the gel-free analysis yields a more comprehensive list of deregulated proteins that are associated with miRNA regulation in general, as well as proteins potentially deregulated by the overexpression of miRNA in the cells. Interestingly, the miR-3687 and miR-4417 overexpression is associated with proteins involved in extracellular endosomes and vesicles.

### 2.4. Cellular Network and Regulatory Pathway Interactions of Deregulated Proteins after miRNAs Transfection

Proteins that were commonly found to be deregulated consistently in the 2D-gel and gel-free analysis (with their expression increasing or decreasing relative to the transfection control) across all four cell lines analyzed and were identified in all three biological replicates were used to build a picture of the cellular regulatory process affected by the miR-3687 and miR-4417 overexpression. The open-source bioinformatic tool Reactome [32] was used to predict the pathway interactions of the proteins regulated by miR-3687 and miR-4417. The deregulated proteins identified after the miR-3687 and miR-4417 overexpression using both 2D and gel-free analysis are listed in Appendix C and Appendix D, respectively.

Both miRNAs shared overlapping regulatory activity, in line with the results of the functional enrichment analysis performed. The list of the most significant regulatory processes and interactions is listed in Table 2. The proteins affected by these miRNAs are involved in different regulatory processes with either miR-3687 or miR-4417 affecting the specific processes. The Reactome analysis performed was not tissue-specific, therefore, did not discriminate if the interactions identified were prostate- or prostate cancer-specific.

Overall, the distinct pathways identified place miR-3687 and miR-4417 activity in key regulatory crossroads (Figure 6). Important regulatory pathways described to be deregulated in prostate cancer could be identified—namely, the metabolism of fatty acids and Rho-GTPase activity. Processes associated with miR activity, such as mRNA splicing and protein transcription, were also identified for miR-3687 and miR-4417. Interestingly, the overexpression of both miRs interferes with the proteins associated with endosome trafficking and gap-junction formation, which can be related to cell-to-cell communication mechanisms.

### 2.5. Western Blot Analysis of Proteins of Interest Deregulated by miR-3687 and miR-4417

To further evaluate the role of miR-3687 and miR-4417 in the regulation of prostate cancer, the expression of proteins of interest that were identified to be involved in the predicted regulatory interactions based on the global proteomics results, were analyzed by Western blotting. Two isoforms of androgen receptor (AR)—high molecular weight (100 kD; AR-100) and low molecular weight (55 kD; AR-50)—were analyzed. AR is the single most important regulatory mechanism in prostate cancer and its functional loss leads to CRPC onset. The expressions of three further proteins were analyzed. Beta-3-tubulin (TBB3), which is involved in microtubule polymerization, affects cytoskeleton organization and intracellular trafficking. Tumor protein D52 isoform 1 (TPD52-IF1) is a prostate-specific and androgen-regulated protein implicated in CRPC progression [33]. The voltage-dependent anion-selective channel 1 (VDAC1) regulates metabolism and apoptotic processes (Figure 7).

#### 2.5.1. miR-3687 Overexpression

AR-100 is not expressed in PC-3 and its expression is not affected in LNCaP. In CTM-LNCaP and VCaP, AR-100 expression is downregulated by 0.6-fold and 0.8-fold, respectively. AR-50 expression is not altered in LNCaP and it is downregulated by 0.6-fold in PC-3 and by 0.7-fold in both CTM-LNCaP and VCaP. TBB3 expression is downregulated by 0.9-fold in PC-3 and CTM-LNCaP, and by 0.8-fold in LNCaP and VCaP cells. TPD52-IF1 is not expressed in PC-3 cells and it is downregulated by 0.3-fold in LNCaP, 0.03-fold in CTM-LNCaP, and 0.6-fold in VCaP. VDAC1 expression is not altered in LNCaP, while it is upregulated by 1.6-fold in PC-3 and 1.4-fold in CTM-LNCaP. In VCaP, VDAC1 expression is downregulated by 0.6-fold. A summary of the Western blot analysis after miR-3687 transfection can be found in Appendix F, Table A9.

#### 2.5.2. miR-4417 Overexpression

AR-100 is not expressed in PC-3 and its expression is upregulated by 1.3-fold LNCaP. In CTM-LNCaP and VCaP, AR-100 expression is downregulated by 0.9-fold and 0.8-fold, respectively. AR-50 expression is downregulated by 0.6-fold in PC-3 and by 0.7-fold in CTM-LNCaP. In both LNCaP and VCaP, AR-50 expression is upregulated by 1.2-fold. TBB3 expression is downregulated by 0.9-fold in PC-3, by 0.4-fold in LNCaP, and by 0.7-fold in both CTM-LNCaP and VCaP cells. TPD52-IF1 is not expressed in PC-3 cells and it is downregulated by 0.6-fold in LNCaP, 0.1-fold in CTM-LNCaP, and by 0.6-fold in VCaP. VDAC1 expression is upregulated by 1.3-fold in PC-3 and 1.4-fold in VCaP. In LNCaP and CTM-LNCaP, VDAC1 expression is downregulated by 0.4-fold and 0.3-fold, respectively. A summary of the Western blot analysis after miR-4417 transfection can be found in Appendix F, Table A10.

## 3. Discussion

The overexpression of miR-3687 and miR-4417 led to a substantial deregulation of several proteins that take part in diverse cellular processes. These changes in protein expression were made evident across the four cell lines used in this study by employing two proteomic quantification methods. Combining two dimensional-gel electrophoresis and mass-spectrometry-based global protein analysis provides a robust analysis. Both analyses are complementary and together can capture a higher number of protein interactions, offering a wider perspective on the deregulated proteome after the overexpression of miRNAs.

Overall, miR-3687 and miR-4417 have significant functional overlap, with several cellular processes affected by both miRNAs. This observation can be partially explained by the fact that miRNA, regardless of function, will impact the cellular machinery involving RNA and protein binding, as well as translation. Other commonly affected pathways can also be the result of the miRNA transfection itself, since the cells express miR levels outside of their normal physiological conditions.

Interestingly, gene ontology analysis potentially shows that both miR-4417 and miR-3687 can be localized in the cytoplasm as well as in endosomes and vesicles trafficked to the extracellular space (Table 1). The activity of myosin, a motor protein that facilitates the mobility of endosomes along cytoskeleton fibers, is also impacted by miR-3687. Therefore, both miRNAs are found in endosomes that are mobile within the cells and are bound to the extracellular space. These findings reiterate the fact that miRNAs are secreted from the prostate, as recently described by Benoist et al., where miR-3687 levels are found elevated in the serum of CRPC patients [34]. Further, our evidence also supports the fact that miRNAs are secreted and enclosed in exosomes in cancer [35].

Reactome analysis identified several regulatory pathways involved in cellular homeostasis, namely, cancer development and progression (Table 2). Given the broad nature of the processes regulated by miR-3687 and miR-4417, our data indicate that these miRNAs support the maintenance of the prostate cancer phenotype by interacting with key signaling pathways. Notably, the involvement of both miRNAs in the metabolic activity of cells and the regulation of the glucose transporter Glut4 (SLC2A4) is trafficked to the membrane, indicating that glycolytic activity is potentially upregulated. The identification of pathways involved in vesicle biogenesis and tracking reiterates the fact that miR-4417 and miR-3687 interact and are transported in endosomes and exosomes. Moreover, the involvement of the miRNAs in RNA and protein metabolism further illustrates the impact of miRNA overexpression on the cellular activity.

Western blot analysis of the expression of proteins identified to be affected by miR-3687 and miR-4417, as well as the androgen receptor, unveiled further differences between the activity of the miRNAs. The prototypical 110 kD AR variant [36], widely known not to be expressed in PC-3 cells [37], is slightly upregulated in LNCaP after miR-4417 transfection. On the other hand, the 50 kD AR splice variant [38] is downregulated in the androgen insensitive cell lines PC-3 and CTM-LNCaP but not in LNCaP and VCaP, which are sensitive to androgens. TBB3, a type of tubulin in which an increased expression is associated with prostate cancer progression [39,40], is not affected by miR-3687 but it is downregulated by miR-4417. The increased AR 110 kD expression and loss of TBB3 expression in LNCaP cells suggest that miR-4417 positively affects these androgen-sensitive cells. The fact that miR-4417 decreases the AR 50 kD expression in androgen-insensitive cells and creates a slight deregulation in TBB3 in androgen-sensitive cells suggests a role in ablating the activity of the AR in CRPC.

TPD52 can be found to be overexpressed in several cancers and, to date, its function is largely unknown [41]. Previously, we reported that the overexpression of TPD52-IF1, a prostate-specific TPD52 isoform, in LNCaP promotes neuroendocrine trans-differentiation (NET), a process believed to be closely tied to the acquisition of the CRPC phenotype [33]. Our results show that TPD52-IF1 is absent in PC-3 and substantially downregulated in CTM-LNCaP after the overexpression of both miR-3687 and miR-4417. miR-3687, which also promoted TPD52-IF1, decreases in LNCaP and VCaP, albeit to a lesser extent. These findings show that these miRNAs promote the loss of TPD52-IF1 in androgen-sensitive cells. Considering that TPD52-IF1 is allegedly key to NET during CRPC onset [42], its loss in androgen-sensitive cells may indicate that NET is not reservable once castration resistance is fully acquired. VDAC1 is expressed in the mitochondria that have also been associated with the development of malignancies, including prostate cancer. This channel ion channel plays a role in both the metabolic shift towards glycolysis and the suppression of apoptosis; therefore, regulating two major cancer pathways [43,44]. miR-4417 downregulated the expression of VDAC1 in LNCaP with a slight upregulation observed in the other cell lines. Taken together, these findings show that key regulators of the prostate cancer phenotype are mainly affected by miR-4417 in LNCaP. This regulatory activity may be due to the androgen-sensitive nature of these cells—an observation compounded by the fact that CTM-LNCaP cells, which are LNCaP that were selected to be androgen resistant, are differentially affected by this miRNA.

In addition to proteomics analysis, prospective miR-3687 and miR-4417 pathophysiological characterization could benefit from the analysis of the transcriptome. Such an analysis, since it can inform about mRNA regulation, would elucidate about the upstream impact of microRNAs, which are the direct targets of miRNA. Using the online miRNA target prediction tool Target Scan [45], we retrieved 644 hits for miR-3687 and 3437 hits for miR-4417. Of these targets, only three genes affected by miR-4417 were commonly identified in our analysis (Appendix G, Table A11). This reflects the fact that the overwhelming majority of identified proteins do not bind the microRNAs, and its deregulation is likely downstream of the direct microRNA interaction. As such, these results highlight the major discrepancies that are often observed when employing different techniques, either analytical or in silico, to investigate target proteins or microRNAs.

Overall, our findings show that miR-3687 and miR-4417 have a wide-ranging effect in the regulation of cancer-specific pathways in prostate cancer, as seen by ontology and reactome analyses (Table 1 and Table 2). Although these analyses could not associate the miRNA with a specific regulatory phenomenon in prostate cancer, cross-analysis of the proteomic results point toward deregulated targets, upstream of the pathways observed. TPD52-IF1 observed the most downregulation by both miRNAs in the CRPC models. Further studies should consider the know-out of this protein in androgen-sensitive cells and evaluate CRPC phenotype progression. Furthermore, the fact that miR-4417 and miR-3687 are associated with endosomes and extracellular vesicles reiterates their use as potential biomarkers [27]. MicroRNAs, in general, have long been speculated to be excreted from cells via exosomes, which require intricate intracellular machinery and vesicular trafficking, to be produced and secreted [46]. Secreted miR-3687 and miR-4417 may fulfill different biological functions, such as cell–cell communication, and support cancer progression by delivering onco-miRNAs to non-malignant cells; at this stage, these functions are still speculative [47,48]. Nonetheless, circulating exosomes loaded with miR-4417 and/or miR-3687 can facilitate their detection in biological fluids and potential use as biomarkers. The value of these miRNAs as biomarkers, for clinical applications or during drug discoveries, is still to be seen and will depend on our further understanding of their biological function and specific role in prostate cancer and CRPC.

## 4. Materials and Methods

### 4.1. Cell Culture

In the present study, four prostate cancer cell lines were used. The androgen-sensitive LNCaP (DSMZ GmbH, Braunschweig, Germany) and VCaP (ATCC, Manassas, VA, USA) cells are representative of prostate adenocarcinoma and the androgen-insensitive PC-3 (ATCC, Manassas, VA, USA) and CTM-LNCaP (subline of LNCaP) cells. To establish CTM-LNCaP, the parental LNCaP cells were cultivated for at least 3 months in phenol red-free RPMI medium supplemented with hormone-free charcoal-treated FBS.

All cells were cultured in RPMI-1640 antibiotic-free medium (Invitrogen, Waltham, MA, USA), supplemented with L-Glutamine, and grown in a 5% CO_2_ atmosphere at 37 °C. PC-3 and VCaP cells were cultured with 10% fetal bovine serum (FBS), CTM-LNCaP with 10% charcoal-treated fetal bovine serum (FBS), and LNCaP cells with 20% FBS.

### 4.2. miRNA-3687 and miR-4417 Transient Transfection

MicroRNA-3687, -4417 and negative control miRNA#2 were obtained as pre-miR^TM^ miRNA Precursor from Ambion (offered by Thermo Fisher Scientific, Waltham, MA, USA). Transient transfection into the selected cell lines was achieved using jetPRIME^®^ reagent (Polyplus, Illkirch, France). Briefly, the transfection solution (200 µL) consisting of microRNA diluted in jetPRIME^®^ buffer and 4 µL jetPRIME^®^ reagent was prepared according to the manufacturers instructions. For detailed information regarding the final concentration, see Appendix E, Table A5. The transfection mix was dropwise added to the cells cultured in 3.5 cm^2^ dishes at 50% confluency. After 3 days, cells were trypsinized and reseeded in a 6 cm dish to cultivate for 3 more days (total cultivation time 6 days). All samples were harvested by TriFast™ reagent (peqGOLD TriFast™, peqlab, Erlangen, Germany) according to the manufacturers instructions. Total RNA was used to assess transfection success by stem-loop qPCR. Protein fraction was used for all proteomic approaches.

### 4.3. Stem-Loop RT-qPCR miRNA Expression Analysis

MicroRNA expression in cell lines was determined by stem-loop RT-qPCR analysis [49]. For the stem-loop RT reaction, TriFast™ extracted total RNA was used. Reverse transcriptase reactions contained 200 ng RNA, 50 nM miR-specific stem-loop RT primer, 50 nM RT-primer for RNU6, 1 × RT buffer (Promega GmbH, Erlangen, Germany), 0.25 mM dNTPs, 75 U M-MLV Reverse transcriptase (Promega GmbH, Erlangen, Germany), and 3.75 U RNase inhibitor (Promega GmbH, Erlangen, Germany) in a total volume of 15 µL. The reactions were incubated in an Eppendorf Mastercycler for 30 min at 16 °C, 30 min at 42 °C, 5 min at 85 °C, and then held at 8 °C. All Reverse transcriptase reactions were filled up to 50 µL with nuclease-free water. The qPCR reaction was carried out with SensiMix™ SYBR^®^ Hi-ROX (Meridian Bioscience, Memphis, TN, USA) using the Bio-Rad CFX96 system. The reaction volume was 15 μL and comprised 7.5 μL of SensiMix, 2.5 μL of diluted cDNA, and 500 nM of each primer. For detailed primer sequence information, see Appendix E, Table A6. The following protocol for the PCR run was used: the polymerase activation (95 °C for 10 min), amplification, and quantification program was repeated 44 times (95 °C for 10 s, 60 °C for 15 s, 72 °C for 5 s with a single fluorescence measurement) in addition to the melting curve program (65–95 °C with a heating rate of 0.5 °C/s and a continuous fluorescence measurement). All samples including blank controls were run in duplicates. Data analysis was performed with the CFX Manager™ software 2.0 (Biorad, Hercules, CA, USA) which employs a ΔΔC(q) relative quantification algorithm and single reference gene (RNU6) normalization.

### 4.4. 2D Gel Electrophoresis

A total protein was isolated from cell lysates, following RNA isolation, by precipitating and removing DNA from the intermediate phase with ethanol followed by centrifugation at 12,000× *g* for 2 min at 4 °C. Proteins in the supernatant were precipitated with isopropanol and centrifuged at 12,000× *g* for 10 min at 4 °C. Protein pellets were washed extensively in 0.3 M guanidinium chloride in 95% (*v*/*v*) ethanol followed by 100% ethanol. Pellets were dried and resuspended in lysis buffer (8 M urea, 2 M thiourea, 4% Chaps, 40 mM Tris-base, and 65 mM DTT). Extracted protein concentrations were determined using the Bradford assay. For 2D gel electrophoresis, isoelectric focusing (IEF) was performed in immobilized pH gradient (IPG) strips with a range of 3–10. Strips were loaded with 150 µg protein topped with 450 µL of rehydration buffer (8 M urea, 2 M thiourea, 2% Chaps, and 50 mM DTT) supplemented with 0.5% IPG buffer (*v*/*v*) and rehydrated overnight. Separation was achieved with an IPGphor system using a pre-programmed voltage gradient (Appendix E, Table A6). Following IEF, IPG strips were equilibrated using two consecutive buffers for 15 min (buffer 1: 6 M Urea, 375 mM Tris HCl (pH 8.8), 20% (*v*/*v*) Glycerol, 1–2% DTT, 2% (*w*/*v*) SDS in dH_2_O; buffer 2 contains 2.5% IAA instead of DTT). Electrophoretic protein separation was performed using 12% SDS-PAGE gels sealed with 1% agarose. Gels were initially run at 0.5 W with power increased to 2 W after 2 h. Afterwards, gels were fixed in 40% methanol and 10% acetic acid and were washed before Coomassie blue staining overnight, and excess background staining was removed with 20% methanol washes.

### 4.5. Mass Spectrometry

Following the analysis and quantification of the 2D gels, differentially expressed protein spots were excised from the gels. Coomassie-stained cut-outs were prepared using an Ettan Spot-handling Platform. Briefly, gels were decolorized in 100 μL NH_4_HCO_3_ in 50% (*v*/*v*) methanol for 30 min and once with 100 mL 75% (*v*/*v*) ACN for 10 min. Following Trypsin incubation, peptides were extracted using 0.1% (*v*/*v*) TFA/50% (*v*/*v*) ACN. The resulting peptide containing supernatant was transferred and the extraction step was repeated. Pooled peptide extracts were dried and dissolved in 0.5% (*v*/*v*) TFA/50% (*v*/*v*) ACN and spotted on the MALDI target (matrix solution: 50% (*v*/*v*) ACN/0.5% (*w*/*v*) TFA saturated with a-cyano-4-hydroxycinnamic acid). The sample measurement was carried out with the 4800 Plus MALDI TOF/TOF™ Instrument. Peptides from cell lysates, without 2D-electrophoretic separation, were isolated by initially reducing a 1 M Urea protein solution with 25 mM DTT in 20 mM NH_4_HCO_3_ followed by alkylation with 100 mM IAA in 20 mM NH_4_HCO_3_. Peptide digestion was performed overnight with trypsin and stopped using 5% (*v*/*v*) CH_3_COOH. Subsequently, peptides were purified using ZipTip^®^ Pipette Tips according to the manufacturer specifications. Finally, samples were eluted twice with 50% and 80% (*v*/*v*) ACN in 1% (*v*/*v*) CH₃COOH and narrowed down to 2 µL. For mass spectrometry, the samples were diluted in 0.05% (*v*/*v*) CH₃COOH/2% (*v*/*v*) ACN to a final volume of 20 µL and analyzed by Proxeon nano-LC system (Proxeon, Odense, Denmark) connected to an LTQ-Orbitrap-MS (ThermoElectron, Bremen, Germany) equipped with a nano-ESI source.

### 4.6. Western Blotting

Electrophoretic protein separation was performed using 12% SDS-PAGE gels loaded with 20 ug of protein at 12–150 V. Subsequently, gels were transferred to a nitrocellulose membrane and were stained with Ponceau-S solution. Membranes were washed and blocked before incubation with the primary antibodies for either 1 h at room temperature or overnight at 4 °C. Secondary antibodies were applied for 30 min and membranes were treated with the enhanced chemiluminescence (ECL) solution. Signals were captured on X-ray films. Protein bands were digitized using a ViewPix 1100 Gel scanner. The quantification was obtained by comparison with the whole protein detected in the Ponceau picture. The conditions and antibodies used for Western blotting analysis are described in Appendix E, Table A8.

### 4.7. Data Analysis

A 2D-gel image analysis was performed using Delta2D Software version 4.6 (DECODON GmbH, Greifswald, Germany) [50]. All gel images were matched and a synthetic fusion gel was prepared for spot detection. A student’s *t*-test was performed to assess the statistical significance of differentially expressed proteins. Based on the average spot volume ratio, spots whose relative expression was changed at least 1.8-fold were considered for further evaluation. The hierarchical clustering and the principal component analysis were performed corresponding to the Delta2D Software protocol. The hierarchical clustering grouped both samples (gel images) and expression profiles. The cluster composition reflected the structure of the experiment, e.g., replicates and images. PCA was performed with the whole set of detected spots of all gel images, resulting in a three-dimensional visualization. The information of all samples was linked together by an orthogonal transformation and an overall pattern was presented in the form of the principal components.

For Maldi-MSMS data, a combined database search of MS and MS/MS measurements was performed using the GPS Explorer software v3.6 (Applied Biosystems, Foster City, CA, USA) [51]. Peak lists were compared with the SwissProt_2014 restricted to human taxonomy using the Mascot search engine 1.9 (Matrix Science Ltd., London, UK). Peptide mixtures that yielded at least twice a morse score of at least 56 were regarded as positive identifications. For protein identification based on LC-MSMS data, automated database searches using the Sequest algorithm rel. 2.7 (Sorcerer built 4.04, Sage-N Research Inc., Milpitas, CA, USA) were performed. The human_uniprot_sprot_2014_02_fwd_rev database was searched assuming the strict trypsin digestion. Sequest search parameters were set to a fragment ion mass tolerance of 1 Da and a parent ion tolerance of 10 ppm. Oxidation of methionine and carbamidomethyl of cysteine was specified in Sequest as variable modifications. Scaffold (version Scaffold_5.1.2, Proteome Software Inc., Portland, OR, USA) was used to validate MS/MS-based peptide and protein identifications. Peptide identifications were accepted if they could be established at greater than 95.0% probability by the Peptide Prophet algorithm with Scaffold delta-mass correction. Protein identifications were accepted if they could be established at greater than 95.0% probability and contained at least 2 identified peptides.

The graphical representation of the data was performed using GraphPad Prism software v. 6.0 (GraphPad Prism software Inc., La Jolla, CA, USA). Data were presented as boxplots with whiskers of min–max and median values for qPCR analysis or as bars with a mean value and standard deviation of the mean for quantitative Western blot analysis. All the experiments were performed in replicates of 3.

The Venn diagrams were created using the software Venny 2.1. (https://bioinfogp.cnb.csic.es/tools/venny/index.html accessed on 21 March 2022).

## 5. Conclusions

With this study, we determined that miRNA-3687 and miR-4417 have a wide range of regulatory functions that maintain the prostate cancer phenotype, from sustaining glycolytic metabolism to inflammation. The overexpression of both miRNAs substantially decreased the expression of TPD52-IF1 in androgen-insensitive cells, indicating that these miRNAs play a key role in regulating NET, leading to CRPC. Moreover, miRNA-3687 and miR-4417 were associated with endosome trafficking and extracellular vesicles, a fact that points towards a secretory mechanism for these miRNAs. This can be exploited to further validate their applications as prostate cancer or CRPC biomarkers.

## Figures and Tables

**Figure 1 ijms-23-10565-f001:**
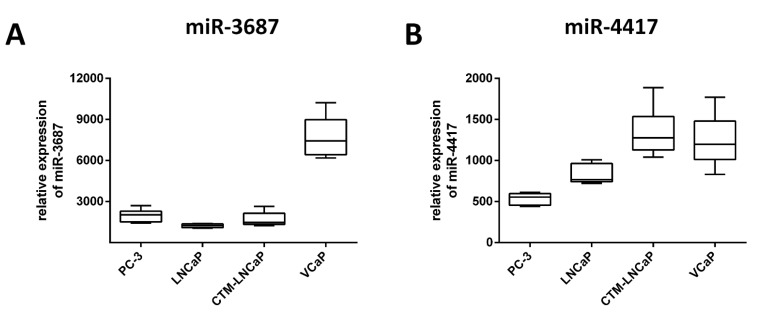
Relative expression of miRNA in prostate cancer cells after transient transfection. The transfection efficacy of miR-3687 (**A**) and miR-4417 (**B**) was analyzed using qPCR. Cultivation time: 6 days; Concentration of miR-3687 for: PC-3 20 nM, LNCaP 40 nM, CTM-LNCaP 40 nM, VCaP 20 nM; Concentration of miR-4417 for: PC-3 20 nM, LNCaP 40 nM, CTM-LNCaP 40 nM, VCaP 20 nM; Concentration of control miR for: PC-3 20 nM, LNCaP 40 nM, CTM-LNCaP 40 nM, VCaP 20 nM.

**Figure 2 ijms-23-10565-f002:**
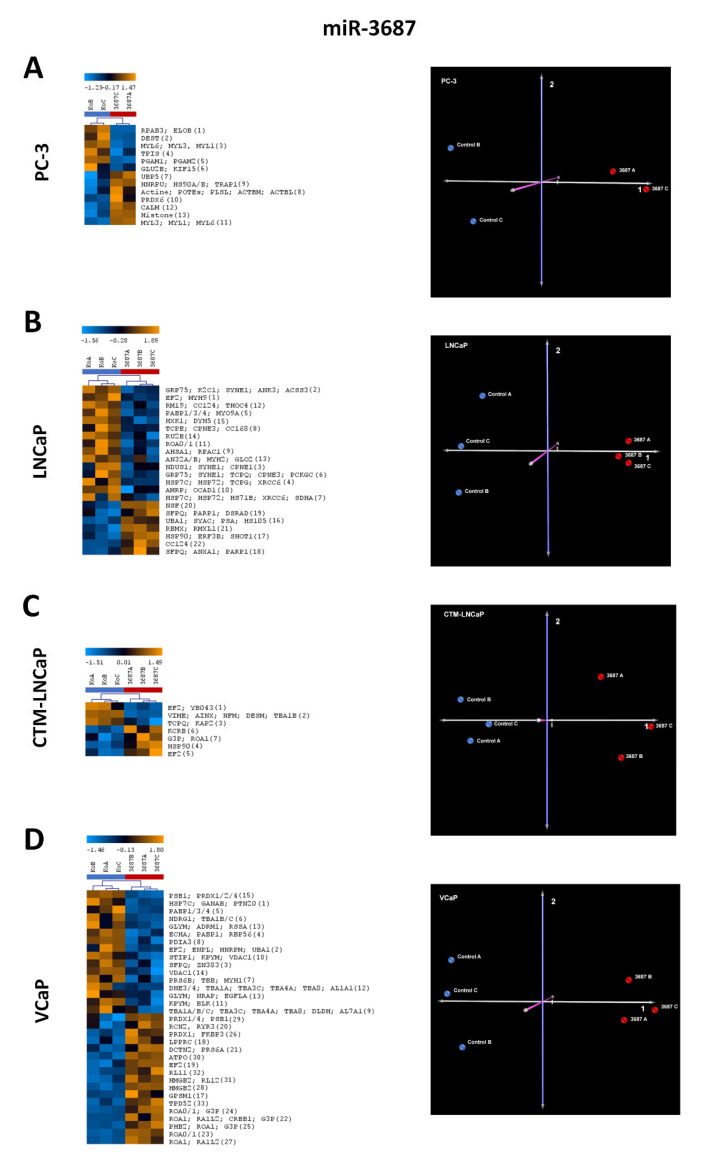
The 2D gel electrophoreses of the four prostate cancer cell line lysates after transfection with miR-3687 and mass spectrometric identification of differentially expressed proteins. (**A**) Results from PC-3 cells with a heatmap of identified proteins (13 spots comprising 27 proteins) in replicates and a principle component analysis (PCA) plot of the samples analyzed. (**B**) Results from LNCaP cells with a heatmap of identified proteins (22 spots comprising 64 proteins) in replicates and a PCA plot of the samples analyzed. (**C**) Results from CTM-LNCaP cells with a heatmap of identified proteins (7 spots comprising 14 proteins) in replicates and a PCA plot of the samples analyzed. (**D**) Results from VCaP cells with a heatmap of identified proteins (33 spots comprising 83 proteins) in replicates and a PCA plot of the samples analyzed. For all cell lines, PCA shows a clear separation between the experimental groups (miR-3687 transfection) and control groups (vector transfection), validating the impact that miRNA transfection has on overall protein expression. The number of spots identified in the 2D gels (Appendix A) is defined in brackets in the heat maps protein list. PCA axis components: 1 (gray), 2 (violet), and 3 (pink).

**Figure 3 ijms-23-10565-f003:**
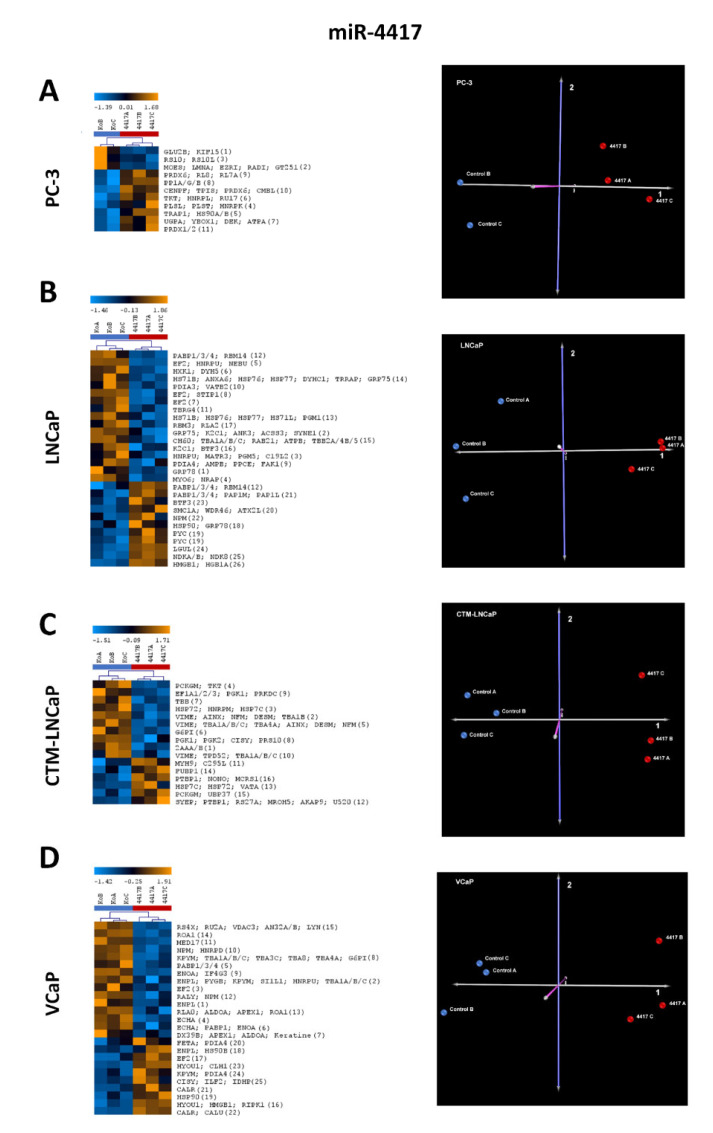
The 2D gel electrophoreses of the four prostate cancer cell line lysates after transfection with miR-4417 and mass spectrometric identification of differentially expressed proteins. (**A**) Results from PC-3 cells with a heatmap of identified proteins (11 spots comprising 34 proteins) in replicates and a principle component analysis (PCA) plot of the samples analyzed. (**B**) Results from LNCaP cells with a heatmap of identified proteins (28 spots comprising 56 proteins) in replicates and a PCA plot of the samples analyzed. (**C**) Results from CTM-LNCaP cells with a heatmap of identified proteins (16 spots comprising 53 proteins) in replicates and a PCA plot of the samples analyzed. (**D**) Results from VCaP cells with a heatmap of identified proteins (25 spots comprising 66 proteins) in replicates and a PCA plot of the samples analyzed. For all cell lines, PCA shows a clear separation between the experimental groups (miR-4417 transfection) and control groups (vector transfection), validating the impact that miRNA transfection has on overall protein expression. The number of spots identified in the 2D gels (Appendix B) is defined in brackets in the heat maps protein list. PCA axis components: 1 (gray), 2 (violet), and 3 (pink).

**Figure 4 ijms-23-10565-f004:**
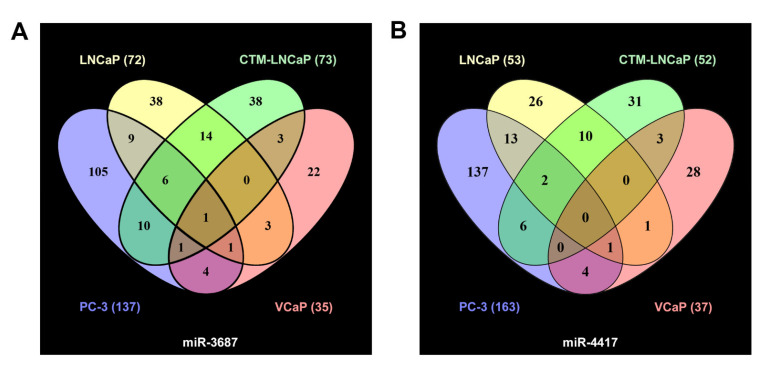
Proteome analysis of four prostate cancer cell lines after mass spectrometric identification—gel-free approach. (**A**) Venn diagram depicting the differentially expressed proteins and their overlap within the four cell lines analyzed after treatment with miR-3687. (**B**) Venn diagram depicting the differentially expressed proteins and their overlap within the four cell lines analyzed after treatment with miR-4417.

**Figure 5 ijms-23-10565-f005:**
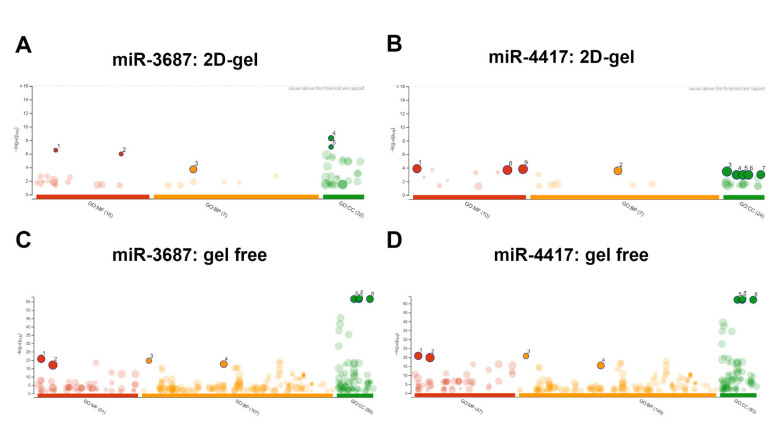
Functional enrichment analysis of differentially expressed proteins after miRNA transfection. (**A**) g:Profiler analysis for miR-3687 based on results from 2D-gel analysis (**B**) g:Profiler analysis for miR-4417 based on results from the 2D-gel analysis. (**C**) g:Profiler analysis for miR-3687 based on results from the gel-free analysis (**D**) g:Profiler analysis for miR-4417 based on results from the gel-free analysis. Figure legend: GO—gene ontology; molecular function (MF); biological pathways (BP); cellular component (CC); dots and numbers: ontology terms of interest (see Table 1).

**Figure 6 ijms-23-10565-f006:**
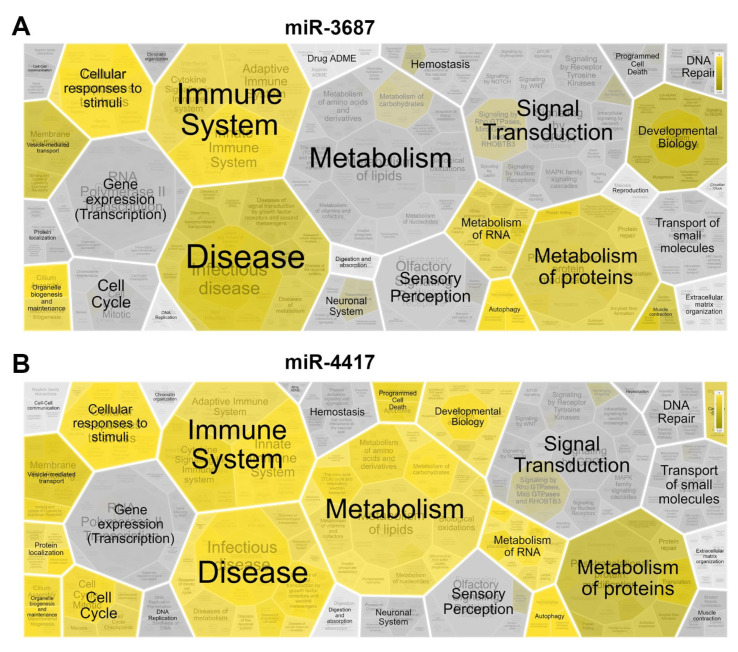
Proteome analysis of the mass spectrometric identification results from the four analyzed cell lines. (**A**,**B**) Reactome network analysis depicted as a hierarchical Voronoi visualization for miR-3687 and miR-4417, respectively.

**Figure 7 ijms-23-10565-f007:**
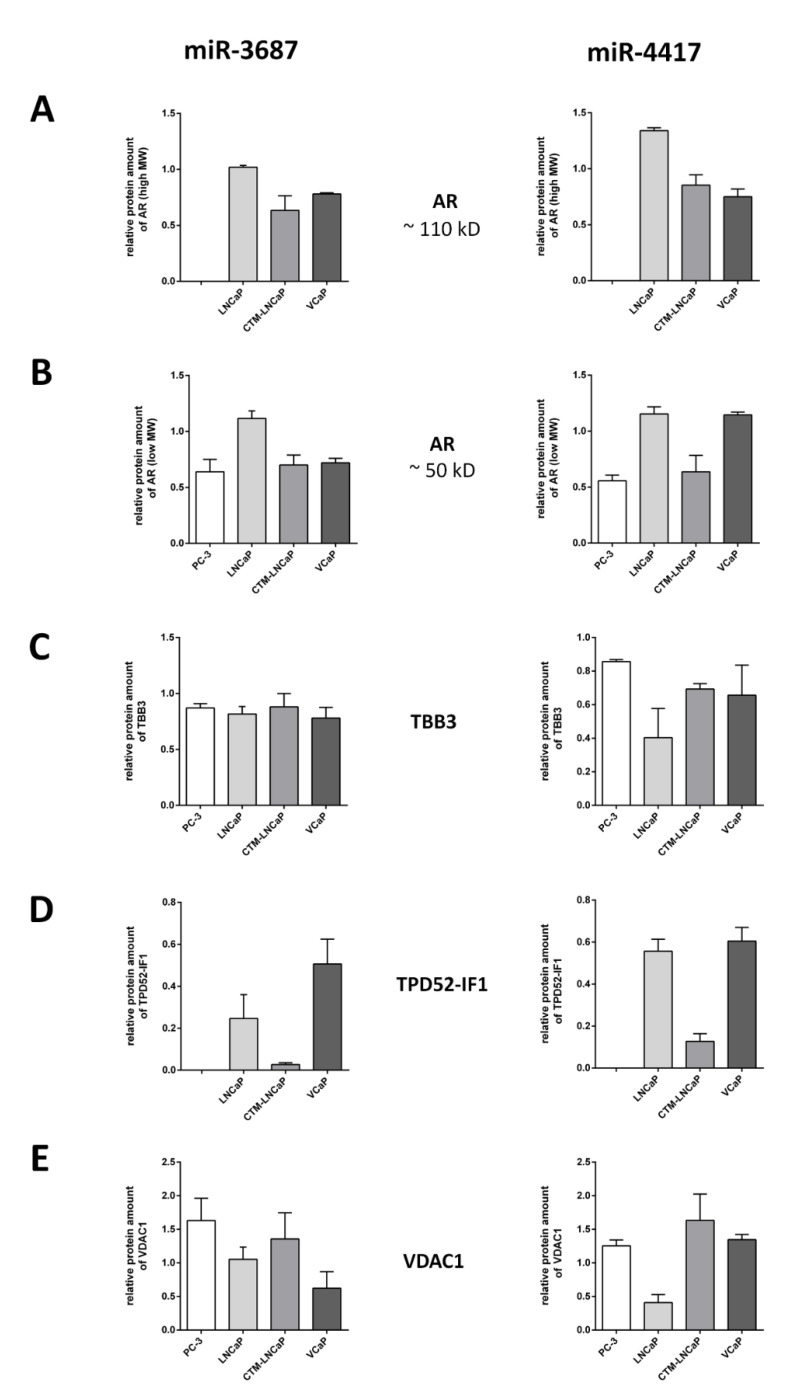
Quantitative Western blot analysis of proteins of interest after transfection with miR-3687 or miR-4417. (**A**) Androgen receptor (AR)—high molecular weight range (110 kD). (**B**) Androgen receptor (AR)—low molecular weight range (50 kD). (**C**) Beta-3-tubulin (TBB3). (**D**) Tumor protein D52 isoform 1 (TPD52-IF1). (**E**) Voltage-dependent anion-selective channel protein 1 (VDAC1). Expression values are depicted relative to transfection controls (1-fold).

**Table 1 ijms-23-10565-t001:** Gene ontologies (GO) identified using g: Profiler functional enrichment analysis. Three ontologies were analyzed: molecular function (MF), biological processes (BP), and cellular compartment (CC). The ontology IDs depicted in Figure 5 are listed in brackets in the corresponding 2D and gel-free analysis.

Source	Term ID	Term Name	miR-3687		miR-4417	
			2D	Gel-Free	2D	Gel-Free
GO:MF	GO:0008379	Thioredoxin activity	X (1)			
GO:MF	GO:00151920	Peroxiredoxin activity	X (2)			
GO:MF	GO: 0003723	RNA binding		X (1)	X (1)	X (1)
GO:MF	GO:0005515	Protein binding		X (2)		X (2)
GO:MF	GO:0097159	Organic cyclic compound binding			X (8)	
GO:MF	GO:1901363	Heterocyclic compound binding			X (9)	
GO:BP	GO0010035	Response to inorganic substance	X (3)			
GO:BP	GO:0002181	Cytoplasmic translation		X (3)		X (3)
GO:BP	GO:0034559	Cellular response to stress			X (2)	
GO:BP	GO:0034645	Cellular macromolecule biosynthetic process		X (4)		X (4)
GO:CC	GO:0016459	Myosin complex	X (4)			
GO:CC	GO:0016460	Myosin II complex	X (5)			
GO:CC	GO: 0005737	Cytoplasm			X (3)	
GO:CC	GO:0031974	Membrane enclosed lumen			X (4)	
GO:CC	GO:0043233	Organelle lumen			X (5)	
GO:CC	GO:0043230	Extracellular organelle		X (5)	X (6)	X (5)
GO:CC	GO:0065010	Extracellular membrane-bounded organelle		X (6)		X (6)
GO:CC	GO:0070062	Extracellular exosome		X (7)		X (7)
GO:CC	GO:1903561	Extracellular vesicle		X (8)	X (7)	X (8)

**Table 2 ijms-23-10565-t002:** Cellular processes and pathway interactions identified using Reactome network analysis. The interactions listed have a score (*p*-value) of < 0.05 and were retrieved using commonly deregulated proteins after 2D-gel and gel-free analysis for either miR-3687 or miR-4417.

Process	Interactions	miR-3687	miR-4417
Metabolism of proteins	Actin and tubulin protein folding	X	
Cellular response to stimuli	HSP90 facilitated steroid hormone receptor-ligand interaction	X	
	ROS detox	X	X
	Unfolded protein response		X
	KEAP-NFE2L2 pathway		X
Signal transduction	Rho GTP activation of ROC-kinases and formins	X	X
	PTK6 regulation of RNA metabolism	X	
	TGF-beta downregulation		X
Autophagy	Chaperone mediated macro-autophagy	X	X
Metabolism	Glycolysis	X	
	Glucose metabolism	X	
	Pyruvate cycle	X	X
	Fatty acid metabolism	X	X
	Triglyceride metabolism		X
Vesicle-mediated transport	Gap-junction trafficking and assembly	X	X
	SLC2A4 trafficking to the plasma membrane	X	X
	COPI-mediated reticulum trafficking	X	X
	Lysosome biogenesis		X
Cell cycle	Centrosome disruption during mitosis	X	
Gene expression	Transcription repression of TFAP2A during differentiation	X	
	TP3 regulation of metabolic genes		X
Metabolism of RNA	mRNA splicing of FGFR2	X	X
	mRNA stabilization of AU-rich elements	X	X
	rRNA processing in the nucleus		X
Programmed cell death	DNA-fragmentation induced apoptosis	X	
	RIPK1-mediated necrosis		X
Organelle biogenesis	Cilia formation	X	X
	Mitochondria biogenesis	X	X
Metabolism of proteins	Methylation	X	X
	Translation elongation	X	
	Co-translational targeting to the membrane		X
	CAP-dependent translation		X
Immune system	IL-4/12/13 signaling	X	X
	IFN alfa/beta induction	X	X
	MHC class II antigen presentation		X
Protein localization	Import to the mitochondria	X	X
Disease	ERBB2 mutations associated with drug resistance	X	X
	Cancer-related EGFR variants signaling		X
DNA repair	Global genome nucleotide excision repair	X	
Transport of small molecules	VLDLR degradation		X

## Data Availability

All data presented in this manuscript can be made available upon request from the corresponding author.

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
