# Peer review of "Identification of the Regulatory Targets of miR-3687 and miR-4417 in Prostate Cancer Cells Using a Proteomics Approach"

_ijms, 2022, doi:10.3390/ijms231810565_

Round 1

Reviewer 1 Report

Venz at al. have investigated miR-3687 and miR-4417 that were consistently upregulated miRNA in CRPC. They transfected miR-3687 or miR-4417 in the prostate cancer cells, and analyzed the transfected cell lysates using a proteomics approach. They identified some regulatory target candidates of miR-3687 and miR-4417 in prostate cancer cells.

Major points

Regrettably, this paper lacks the association of both miR-3687 and miR-4417 with their biological functions/evidences and specific phenomena (e.g. cell growth, metabolisms and resistance to apoptosis) in prostate cancer.

Minor comments

Materials and Methods: How to verify whether stem-loop RT-qPCR analysis is affected by contaminants of miRNA-3687 and miRNA-4417 in culture medium?

Results: Page 3, Figure 1. How many % of cells incorporated miRNA-3687 and miRNA-4417?

Author Response

Reply to Reviewer 1 comments

The authors are much appreciated for the constructive feedback to our manuscripts: ´´Identification of the Regulatory Targets of miR-3687 and miR-4417 in Prostate Cancer Cells Using a Proteomics Approach´´. Here we provide a point-by-point reply to the Reviewer comments.

Comments and Suggestions for Authors

Venz et al. have investigated miR-3687 and miR-4417 that were consistently upregulated miRNAs in CRPC. They transfected miR-3687 or miR-4417 in the prostate cancer cells, and analyzed the transfected cell lysates using a proteomics approach. They identified some regulatory target candidates of miR-3687 and miR-4417 in prostate cancer cells.

Major points

  1. Regrettably, this paper lacks the association of both miR-3687 and miR-4417 with their biological functions/evidences and specific phenomena (e.g. cell growth, metabolisms and resistance to apoptosis) in prostate cancer.

RE: The authors strongly agree that experimental studies associating the expression of both miR-3687 and miR-4417 with their biological function would broaden the findings presented in this study and expand our understanding of the mechanisms governed by these microRNAs. Ongoing experimental work in our research groups is aiming at further exploring the pathophysiology of miR-3687 and miR-4417, building on the results presented in this manuscript. In our manuscript we present the results of a proteomics approach in a comprehensive fashion, to the best of our knowledge. The authors believe that further results from the functional characterization studies detailing the biological activity of these microRNAs, with a focus on energy consumption, cell survival, growth and trans-differentiation will bring sufficient scientific novelty to warrant an original publication.

  1. Materials and Methods: How to verify whether stem-loop RT-qPCR analysis is affected by contaminants of miRNA-3687 and miRNA-4417 in culture medium?

RE: We appreciate this observation and acknowledge that miRNA contaminations can be present in the culture medium, deriving from the addition of serum. As described in the methods section (4.2), the cells were transfected only once and kept in culture over a period of 6 days that include passaging and washing. Prior to harvesting, cells were again washed with PBS to remove media traces. Given the high miR-3687 and miR-4417 expression we assume that this is solely due to the transfection and the contribution from possible contaminations appears negligible. 

  1. Results: Page 3, Figure 1. How many % of cells incorporated miRNA-3687 and miRNA-4417?

RE:  The experimental methods used to overexpress miRNA-3687 and miRNA-4417 did not contemplated a fluorescently labeled control miRNA to estimate the transfer efficiency, and unfortunately, we are not able to comment on the percentage of transfected cells. On the other hand, we have extensively used the jetPRIME transfection procedure with consistent results. The subsequent miRNA quantification in all experimental replicates consistently yielded high expression rates (miRNA-3687 > 1500-fold and miRNA-4417 > 500-fold relative to transfection controls). Hence, we assumed a normal distribution within the cell population and negligible effects caused by individual cells with an extremely high miRNA copy number.

Reviewer 2 Report

In the article, “Identification of the Regulatory Targets of miR-3687 and miR- 4417 in Prostate Cancer Cells Using a Proteomics Approach” the authors explore the potential role of hsa-miR-3687, and hsa-miR-4417 in prostate cancer and castration resistant prostate cancer. The laboratory methods used in exploring this topic are interesting; the role of both of this microRNAs in cancer is certainly worth study. Before this manuscript can be published however, a few things must be addressed.

Minor Comments:

1.     Abstract: line 14. Sentence starting “The expression on many miRNAs…” – I believe the word should be “of” rather than “on”.

2.     Abstract: starting line 17 and continuing throughout – This may be unique to this journal but it is confusing to refer to the microRNAs using the truncated abbreviation “miR-XXX” instead of indicating whether the microRNA in question is human in origin (using “hsa-miR-XXX”) or for instance mouse (“mmu-miR-XXX”).

3.     Abstract: line 18 – Sentence starting “PCa and CRPC cell lines…” Correct the verb “overexpressed” to “overexpress”.

4.     Abstract: line 20 – Sentence starting “Whole cell lysate…” There’s a parenthesis closing out the sentence that I cannot follow.

5.     Introduction: lines 58-61. Sentences starting “Non-coding RNAs participate…” This is a small thing but I think these sentences are really unnecessary. The paragraph effectively starts at line 62 with the sentence “MicroRNAs (miRNAs) are small non-coding…. These preceding sentences make the paragraph feel stylistically like it has two starts to it.

6.     Introduction: line 77 Sentence starting “Therefore, miRNAs activity is …”. Please consider rewriting this sentence. It is awkward.

7.      Section “Profiling of differentially expressed proteins after miRNA transfection” - Line 177. Please include a citation describing the paper, rather than the link for this software. Just including the link leaves the reader wondering which version of the webserver was used.

8.     Section Cellular network and regulatory pathway interactions of deregulated proteins after miRNAs 225 transfection” – Line 231 – Please a citation describing the paper for Reactome rather than the link. Multiple versions exist of its knowledgebase and distinction is needed.

9.     Methods – line 389 – Sentence starting – “In the present study …” Were these cells purchased? I do not believe I could find any description on their origin.

10.  Methods – line 396 – Sentence starting – “MicroRNA-3687 and -4417…” What was the nature of the control miRNA provided by Thermofisher? How similar is it to miRNA-3687 and -4417?

11.  Table 1 “Primer sequences for qPCR”- Should sequence be spelled “sequenz”?

Author Response

Reply to Reviewer 2 comments

The authors are much appreciated for the constructive feedback to our manuscripts: ´´Identification of the Regulatory Targets of miR-3687 and miR-4417 in Prostate Cancer Cells Using a Proteomics Approach´´. Here we provide a point-by-point reply to the Reviewer comments.

Comments and Suggestions for Authors

In the article, “Identification of the Regulatory Targets of miR-3687 and miR- 4417 in Prostate Cancer Cells Using a Proteomics Approach” the authors explore the potential role of hsa-miR-3687, and hsa-miR-4417 in prostate cancer and castration resistant prostate cancer. The laboratory methods used in exploring this topic are interesting; the role of both of this microRNAs in cancer is certainly worth study. Before this manuscript can be published however, a few things must be addressed.

Minor Comments:

  1. Abstract: line 14. Sentence starting “The expression on many miRNAs…” – I believe the word should be “of” rather than “on”.

RE: We thank the reviewer for spotting this inconsistency. The revised version of our manuscript is now corrected. 

  1. Abstract: starting line 17 and continuing throughout – This may be unique to this journal but it is confusing to refer to the microRNAs using the truncated abbreviation “miR-XXX” instead of indicating whether the microRNA in question is human in origin (using “hsa-miR-XXX”) or for instance mouse (“mmu-miR-XXX”).

RE: In order to clarify the fact that the microRNAs used in this study are indeed human in origin, the initial statement referring to both microRNAs have been changed to: ´´ In this study we used different proteomics tools to analyze the activity of hsa-miR-3687-3p (miR-3687) and hsa-miR-4417-3p (miR 4417), two miRNAs upregulated in CRPC.” The subsequent text the simplified notations miR-3687 and miR-4417 (in brackets) are used.

  1. Abstract: line 18 – Sentence starting “PCa and CRPC cell lines…” Correct the verb “overexpressed” to “overexpress”.

RE: This misconstruction is now corrected in the revised version of our manuscript.

  1. Abstract: line 20 – Sentence starting “Whole cell lysate…” There’s a parenthesis closing out the sentence that I cannot follow.

RE: This typing error is now corrected in the revised version of our manuscript.

  1. Introduction: lines 58-61. Sentences starting “Non-coding RNAs participate…” This is a small thing but I think these sentences are really unnecessary. The paragraph effectively starts at line 62 with the sentence “MicroRNAs (miRNAs) are small non-coding…. These preceding sentences make the paragraph feel stylistically like it has two starts to it.

RE: We thank the reviewer for highlighting this textual misconception. The revised manuscript is now changed so the structure is consistent and reads as follows: ´´MicroRNAs (miRNAs) are small non-coding, single-stranded RNA molecules usually consisting of 17 to 25 nucleotides in length that regulate gene expression at the post-transcriptional level. The expression of these oligonucleotides is extensively reported across several cancers and reported to be involved in tumor pathogenesis and progression“. Literature references have also been updated to maintain the consistency of the text and support our statements accordingly.

  1. Introduction: line 77 Sentence starting “Therefore, miRNAs activity is …”. Please consider rewriting this sentence. It is awkward.

RE: The sentence in question has been rewritten to improve the clarity of our message: “MiRNAs can act as oncogenes (oncomiRs) or as tumor suppressors depending on cell type specific expression and activity´´.

  1. Section “Profiling of differentially expressed proteins after miRNA transfection” - Line 177. Please include a citation describing the paper, rather than the link for this software. Just including the link leaves the reader wondering which version of the webserver was used.

RE: We thank the reviewer for this observation and we have now included the appropriate citation in our revised manuscript referencing the g:Profiler webtool: ´´g:Profiler: a web server for functional enrichment analysis and conversions of gene lists (2019 update) Nucleic Acids Research 2019; doi:10.1093/nar/gkz369´´

  1. Section “Cellular network and regulatory pathway interactions of deregulated proteins after miRNAs 225 transfection” – Line 231 – Please a citation describing the paper for Reactome rather than the link. Multiple versions exist of its knowledgebase and distinction is needed.

RE: We thank the reviewer for this observation and we have now included the appropriate citation in our revised manuscript referencing the Reactome analysis tool: ´´ The reactome pathway knowledge base 2022, Nucleic Acids Research, 2021; https://doi.org/10.1093/nar/gkab1028´´

  1. Methods – line 389 – Sentence starting – “In the present study …” Were these cells purchased? I do not believe I could find any description on their origin.

RE: The information about the cell supplier has been added to the method section (section 4.1) of our revised manuscript.

  1. Methods – line 396 – Sentence starting – “MicroRNA-3687 and -4417…” What was the nature of the control miRNA provided by Thermofisher? How similar is it to miRNA-3687 and -4417?

RE: We appreciate the reviewer for point out this missing information. The details of the MicroRNA-3687 and MicroRNA-4417 sequences has been added to our revised manuscript. The supplier did not provide details regarding the sequence of the control-miR and regrettable we cannot provide a comparison. Moreover, the Blastn-tool (NCBI BLAST®) revealed no significant similarity between the miR-3687 and miR-4417. Both microRNAs differ in length and share an identical sequence of 6 nucleotides (miR-3687: cccggacaggcguucgugcgacgu; miR-4417: ggugggcuucccggaggg)

  1. Table 1 “Primer sequences for qPCR”- Should sequence be spelled “sequenz”?

RE: We have corrected this spelling mistake in our revised manuscript. 

Reviewer 3 Report

In this manuscript, the authors describe the identification of two castration-resistant prostate cancer (CRPC)-associated microRNA targets using proteomics. Using classical 2D-PAGE proteomics combined with gel-free proteomics, the authors identify several targets that are commonly regulated by the two miRNAs, miR-3687 and miR-4417. Those targets reveal molecular changes that contribute to prostate carcinogenesis, which could ultimately lead to the development of new biomarkers and/or drugs to fight advanced prostate cancer.

The approach taken by the authors is sound, using miRNA mimic-transfection of PCa and CRPC cell line models, followed by MS analysis. However, there are some major issues that need to be resolved:

1.     How high is the endogenous expression of the two miRNAs in the cell line models. Wouldn't knockdown of the miRNAs be a better model to study their function?

2.     The presentation of the data is quite confusing. Different pathway plots and gene lists are shown in the paper. It is rather unclear whether they come from the 2D-gel, gel-free, or common (overlapping) datasets. I suggest putting the data from 2D-gel and gel-free proteomics in the supplement. Only put tables and figures containing gene lists and pathways of the overlapping data in the main article. Such an overlapping gene list also should be the basis for selecting candidates for validation by western blotting.

3.     It is unclear where the selection of targets for western blotting comes from. Are these the most consistent targets (in most cell lines and by both miRNAs)? Why has SLC2A4 (aka GLUT4, use official HUGO nomenclature!) not been validated?

4.     Since the main targets of miRNAs are RNA transcript, the paper would benefit from a transcriptome analysis. At the very least, an in silico analysis is desirable. Do the target genes identified here also contain miRNA-binding sequences in their 3'UTR? Show that the miRNAs inhibit the 3'UTR of one or a few targets using reporter assays.

5.     What is the target gene expression of e.g. TPD52 and GLUT4 in clinical samples? You would expect the targets to be down in CRPC, as the miRNAs are upregulated in CRPC. This information is essential to demonstrate clinical relevance of the findings.

Minor issues:

1.     In the legend of figure 1, indicate the miRNA concentrations used and the duration of transfection.

2.     Move the 2D gels to the supplement and explain in the legend what the dots and blue and orange shades mean. In Figures 2 and 3, show only the heatmaps and PCA plots (with white backgrounds !) and explain the purple lines in the PCA plots.

3.     What do all the dots and numbers in Figure 4 (C-F) mean?

4.     In Figure 6, explain the abbreviation “ANDR” on the y-axis of the first four graphs.

5.     In a recent study (PMID 32408351), levels of miR-3687 were shown to be elevated in whole blood from CRPC patients compared to controls. This may support the hypothesis that this miRNA is present in circulating vesicles that are secreted by the tumor.

6.     There is a lot of repetition in the text and the text contains numerous spelling and grammar mistakes. The text therefore needs to undergo a very thorough revision.

Author Response

Reply to Reviewer 3 comments

The authors are much appreciated for the constructive feedback to our manuscripts: ´´Identification of the Regulatory Targets of miR-3687 and miR-4417 in Prostate Cancer Cells Using a Proteomics Approach´´. We have taken the reviewer´s suggestions in consideration to improve our manuscript and here we provide a point-by-point reply to the Reviewer comments.

Comments and Suggestions for Authors

In this manuscript, the authors describe the identification of two castration-resistant prostate cancer (CRPC)-associated microRNA targets using proteomics. Using classical 2D-PAGE proteomics combined with gel-free proteomics, the authors identify several targets that are commonly regulated by the two miRNAs, miR-3687 and miR-4417. Those targets reveal molecular changes that contribute to prostate carcinogenesis, which could ultimately lead to the development of new biomarkers and/or drugs to fight advanced prostate cancer.

The approach taken by the authors is sound, using miRNA mimic-transfection of PCa and CRPC cell line models, followed by MS analysis. However, there are some major issues that need to be resolved:

  1. How high is the endogenous expression of the two miRNAs in the cell line models. Wouldn't knockdown of the miRNAs be a better model to study their function?

RE: In our study the endogenous expression of miR-3687 and miR-4417 was not quantified in absolute terms. In the transfection control samples, the Cq-values for both miRNAs ranged 28-30, indicative of the low expression of these miRNAs in cell lines selected for our study. In this context, it is our understanding that knockdown of miRNA is a more sensible approach when miRNAs are highly expressed, or strongly upregulated, in the cells. Mass spectrometry (MS) analyses revealed expresses changes in the proteome upon miRNA overexpression, with up- and down- regulation of various targets. Given the data obtained from the MS studies it was considered that our overexpression approach was valid to investigate the regulatory targets of miR-3687 and miR-4417, nonetheless, we assume that the use of anti-miRNAs can would also yield changes in proteome. Moreover, using stem-loop qPCR we could quantify the potency of miRNA transfection, as depicted in Figure 1, showing the relative expression after transfection compared to control transfection.

  1. The presentation of the data is quite confusing. Different pathway plots and gene lists are shown in the paper. It is rather unclear whether they come from the 2D-gel, gel-free, or common (overlapping) datasets. I suggest putting the data from 2D-gel and gel-free proteomics in the supplement. Only put tables and figures containing gene lists and pathways of the overlapping data in the main article. Such an overlapping gene list also should be the basis for selecting candidates for validation by western blotting.

RE: We appreciate the remarks regarding the data presentation and have adapted our revised manuscript in order to improve readability. To emphasize the nature of the data presented the headings of the results are more explicit. The different proteomic methods are presented separately at first. Section 2.2.1/2.2.2 contains the results of the gel-based method and section 2.2.3/2.2.4 the results of the gel-free analysis.

The data in Appendix C and Appendix D contains cell line specific quantitative data from both analytical methods. This long list of protein was not included in the main text per se, considering the descriptive nature of the results and the limited overlap between sections.  The two proteomic analysis employed differ in methodology, hence the changes in protein expressions retrieve from both techniques were not expected to be necessarily the same, which is readily apparent from the expression factors. The Scaffold factor refers to the gel-free analysis and the Delta 2D factor to the gel-based method. The determination of a cut-off for further containment must be method specific, so that one cannot perform a common analysis here. For this reason, protein profiling was performed separately for each of the methods used. Our analysis prioritized the comparison of cell lines as a basis for selecting proteins for verification. With miR-3687, only 1 protein (VDAC1) was differentially expressed in all 4 cell lines. This protein was examined by Western blot analyses. The remaining proteins examined overlap in certain conditions tested but not across all cell lines and their selection was based on their potential functional influence in CRPC evolution considering previous reports in literature. This point is now clearly made in the discussion of our revised manuscript and supported by the appropriate literature.

In our revised manuscript, the order and labeling of the Appendix information has been updated to improve the data presentation and avoid any confusion in the interpretation. Every appendix section (A-G) corresponds to a separate set of results.

  1. It is unclear where the selection of targets for western blotting comes from. Are these the most consistent targets (in most cell lines and by both miRNAs)? Why has SLC2A4 (aka GLUT4, use official HUGO nomenclature!) not been validated?

RE: To clarify this comment, we refer point 2 and the fact that selection of candidate targets was based on the influence of development to CRPC and the changes within all cell lines. More extensive validation analyses were not in the scope of this project. We acknowledge the concern the mishap in referring to GLUT2 and it is now designated SLC2A4, with a further clarification of this function. 

Changes in SLC2A4 expression were not determined in the proteomics analyses performed, hence miR-3687 and miR-4417 seemingly to not affect the expression of this glucose transporter. Reactome analysis flagged a potential role for the miRNAs in the downstream regulation of SLC2A4, impacting its membrane localization (trafficking from endosomes to the cell membrane) and not necessarily its expression at the protein level. Considering this fact, SLC2A4 was not a focus of our validation. Further studies can expand on the protein expression levels and incorporate cellular localization analysis of prospective targets (e.g. immunofluorescent co-localization).

  1. Since the main targets of miRNAs are RNA transcript, the paper would benefit from a transcriptome analysis. At the very least, an in silico analysis is desirable. Do the target genes identified here also contain miRNA-binding sequences in their 3'UTR? Show that the miRNAs inhibit the 3'UTR of one or a few targets using reporter assays.

RE: We appreciate this suggestion and do agree that a transcriptome analysis would be valuable to further characterize the targets of these microRNAs. The authors believe that such analysis could in itself warrant an original publication. In addition to our revisions we performed a miRNA target prediction analysis using the online tool target scan (https://www.targetscan.org/vert_80/) The results obtained discussed in the revised version of our manuscript. Overall, target scan retrieved 644 hits for miR-3687 and -3437 hits for miR-4417. Only 3 targets for miR-4417 were also identified in our analysis. Our proteomics analysis identified targets that are affected by miR-3687 and miR-4417 overexpression. The majority of identified proteins do not necessarily bind the microRNAs and its deregulation expression likely to be downstream of direct microRNAs interaction. As such, these results highlight the major discrepancies that are often observed when employing different techniques, either analytical or in silico to investigate target proteins or microRNAs. 

              We also recognize that describing the miRNA binding sequences in the untranslated regions of the mRNA of the identified protein would further validate the miRNA interactions. However, this analysis was beyond the scope of our project. Nonetheless, prospective studies to explore the functional activity of miR-3687 and miR-4417 could incorporate such analysis.

  1. What is the target gene expression of e.g. TPD52 and GLUT4 in clinical samples? You would expect the targets to be down in CRPC, as the miRNAs are upregulated in CRPC. This information is essential to demonstrate clinical relevance of the findings.

RE: In previous studies using clinical samples the expression of TPD52 was determined to be elevated in prostate cancer specimens relative to benign prostate tissue, at the mRNA and protein level (doi:10.1111/j.1742-4658.2008. 06697.x). In this analysis CRPC clinical samples were not included. Further, in the same study overexpression of TPD52 in LNCap cell resulted in enhanced migration while TPD52 resulted in an increase in apoptosis. We have analyzed TPD52 isoform1, a  prostate-specific TPD52 isoform (https://doi.org/10.1158/0008-5472.CAN-03-3331) believed to be involved in trans-differentiation and CRPC progression (https://doi.org/10.1007/s13277-016-4925-1). The upregulation of miR-3687 and miR-4417 in clinical samples is likely related to the regulation of CRPC progression by TPD52. In the scope of this study we have also analyzed also TPD52-isoform 3 (TPD52-IF3) but no changes in expression across all cell lines were observed after miRNAs overexpression. This data is not included in our manuscript.

GLUT4 is characterized to be expressed in the prostate (10.1126/science.1260419) and in both LNCap and PC3 cells (https://doi.org/10.1210/en.2014-1260) at the gene and protein levels. To our knowledge the expression of GLUT4 in clinical samples of prostate cancer or CRPC is unreported.

Minor issues:

  1. In the legend of figure 1, indicate the miRNA concentrations used and the duration of transfection.

RE: This information is now added to Figure 1

  1. Move the 2D gels to the supplement and explain in the legend what the dots and blue and orange shades mean. In Figures 2 and 3, show only the heatmaps and PCA plots (with white backgrounds!) and explain the purple lines in the PCA plots.

RE: As recommended by the reviewer, the 2D gels are presented in appendix material of the revised version of our manuscript. The heatmaps and PCA plots have been recompiled. The background in the PCA plots given by Delta 2D software is specific therefore has not been changed. The colors of the lines corresponding to the axes of the 3D plot representing the components 1 (gray), 2 (purple) and 3 (pink) of the PCA analysis. This information is now added to our revised manuscript.

  1. What do all the dots and numbers in Figure 4 (C-F) mean?

RE: The plots depicted in Figure 4 have been updated to the new Figure 5 (A-D) in our revised manuscript. The information regarding the meaning of all the dots and numbers in the functional enrichment analysis is now added to Table 1, which lists the gene ontologies (GO) identified using the g:Profiler tool.

  1. In Figure 6, explain the abbreviation “ANDR” on the y-axis of the first four graphs.

RE: We have used the Uniprot taxonomy for the designation of the proteins, where ANDR is given for the androgen receptor. We have standardized the nomenclature in our revised manuscript. According to the new figure layout, these results are now present in the new Figure 7.

  1. In a recent study (PMID 32408351), levels of miR-3687 were shown to be elevated in whole blood from CRPC patients compared to controls. This may support the hypothesis that this miRNA is present in circulating vesicles that are secreted by the tumor.

RE: We thank the reviewer for point out this recent study, which we found insightful and our revised manuscript now discussed the implications of this study in light of our results. 

  1. There is a lot of repetition in the text and the text contains numerous spelling and grammar mistakes. The text therefore needs to undergo a very thorough revision.

RE: Our revised manuscript has been thoroughly checked for typos, mistakes and grammatical misconstructions in order to improve readability. Extensive repetitions in the text have been avoided unless deemed necessary to discuss a result or reiterate a conclusion drawn from the data.

Round 2

Reviewer 1 Report

The authors present data demonstrating that miR-3687 and miR-4417 are consistently upregulated in CRPC. They transfected miR-3687 or miR-4417 in the prostate cancer cells, and analyzed the transfected cell lysates using a proteomics approach. They identified some regulatory target candidates of miR-3687 and miR-4417 in prostate cancer cells.

Major points

Although this manuscript demonstrates a large amount of data, the main issue with this study is the lack of biological significance of the findings. This paper falls short of International Journal of Molecular Science standard. More mechanistic insights into the role of miR-3687 and miR-4417 are required to warrant publication in International Journal of Molecular Science.

Author Response

Reply to Reviewer 1 comments (2nd round)

Comments and Suggestions for Authors

The authors present data demonstrating that miR-3687 and miR-4417 are consistently upregulated in CRPC. They transfected miR-3687 or miR-4417 in the prostate cancer cells, and analyzed the transfected cell lysates using a proteomics approach. They identified some regulatory target candidates of miR-3687 and miR-4417 in prostate cancer cells.

Major points

  1. Although this manuscript demonstrates a large amount of data, the main issue with this study is the lack of biological significance of the findings. This paper falls short of International Journal of Molecular Sciencestandard. More mechanistic insights into the role of miR-3687 and miR-4417 are required to warrant publication in International Journal of Molecular Science.

RE: We acknowledge the reviewers concerns with the lack of functional characterization and mechanistic insights on our findings. The authors can reiterate that, retrospectively, previous studies, including our own, have shed light on the biological activity of certain targets identified here for miR-3687 or miR-4417. Namely, the expression of TPD52 was determined to be elevated in prostate cancer clinical specimens relative to benign prostate tissue, at the mRNA and protein level (doi:10.1111/j.1742-4658.2008. 06697.x). Further, in the same study overexpression of TPD52 in LNCap cell resulted in enhanced migration while TPD52 resulted in an increase in apoptosis. We have analyzed TPD52 isoform1, a  prostate-specific TPD52 isoform (https://doi.org/10.1158/0008-5472.CAN-03-3331) believed to be involved in trans-differentiation and CRPC progression (https://doi.org/10.1007/s13277-016-4925-1).

Reviewer 3 Report

Based on my previous comment (no. 4) about the use of 'ANDR' on the y-axis of figure 6, the authors have decided to change AR to ANDR throughout the manuscript. ANDR is a rather unusual, non-HUGO, abbreviation for the androgen receptor gene and protein.

I strongly recommend to use AR (gene/mRNA) and AR (protein) instead !!!

Author Response

Reviewer 3, 2nd round

Comments and Suggestions for Authors:

Based on my previous comment (no. 4) about the use of 'ANDR' on the y-axis of figure 6, the authors have decided to change AR to ANDR throughout the manuscript. ANDR is a rather unusual, non-HUGO, abbreviation for the androgen receptor gene and protein.

I strongly recommend to use AR (gene/mRNA) and AR (protein) instead!!!

Reply: We once again thank the reviewer for pointing out this inconsistency in our manuscript. In the revised version we have corrected ANDR to AR in the text, appendix material, and figure nr. 7
